# Deep Feature Fusion Framework for Alzheimer's Disease Staging using Neuroimaging Modalities

No Author Given

Anonymous

**Abstract.** Alzheimer's Disease (AD) is a significant neurodegenerative disorder. Detecting AD early is essential for effective management and improving the quality of life for both patients and their families. Recent advancements in medical imaging technology have introduced neuroimaging-based methods for early AD diagnosis. However, the challenges in early AD detection suggest that using a single modality dataset in deep learning (DL) studies, particularly neuroimaging, might not yield precise predictions of AD progression compared to integrating data from multiple imaging modalities. Utilizing information from multi-modal data fusion can enhance the detection of subtle changes and biomarkers, leading to more reliable and accurate diagnosis. In our study, we develop an automated multimodal system that integrates MRI and PET images at an intermediate fusion level, facilitating the early diagnosis of Alzheimer's disease. This fusion approach does not require extensive preprocessing steps typically associated with image fusion method. Our proposed methodology outperforms previous studies in differentiating between individuals with Alzheimer's disease and cognitively normal (CN) individuals, achieving an AUC score of 97.67% and an accuracy (ACC) of 95.24%.

**Keywords:** Alzheimer's Disease · Neuroimaging Features · 3D Image Classification.

## 1 Introduction

Alzheimer's Disease (AD) is a severe neurodegenerative disease. Early identification of AD is crucial for effective management and for enhancing the quality of life for both patients and their families. Unfortunately, the majority of existing diagnostic techniques rely on subjective assessments of behavioral and cognitive symptoms, leading to potential unreliability and misdiagnosis. In recent years, advances in medical imaging technology have led to the emergence of neuroimaging- based methods for the early diagnosis of AD. However, these methods often rely on the analysis of a single modality, which may not capture the full complexity of the disease. Multimodal data fusion has been proposed as a promising approach to address this limitation by combining information from different modalities.

In a clinical setting, AD is typically diagnosed by systematically examining various aspects of patients' multiple modalities [18]. These aspects commonly

derive from the diverse information sources of patients, encompassing neuroimage data, gene sequence data, profile data, and clinical mental state scale data. In contrast to the classification of AD based solely on single-modal neuroimages, enhanced performance can be attained through the utilization of multi-modal classification, involving the integration of diverse information sources. Investigating the synergies among various multi-modal neuroimages significantly contributes to the identification of pathological processes in neurological disorders. This technique has found applications in image classification [20, 23] and image registration [8]. The motivation for engaging in multi-modal fusion stems from two primary advantages: firstly, the potential for more robust predictions through the observation of the same phenomenon across multiple modalities [4], and secondly, the extraction of complementary information from diverse modalities to enhance the precision of classification results [3].

The multimodal framework consists of essential components that are primarily structured in three key levels. The initial level, known as the integration level, involves the definition of various modalities of data intended for fusion. Thus, at this stage, the determination is made regarding what specifically should be fused. The subsequent level is the fusion methodology, encompassing the approach employed to combine the identified data, guided by the chosen fusion strategy. In the literature, fusion strategies are classified into three groups: Early fusion, also known as feature-level fusion, is the process of merging multimodal data by concatenating its features in a vector, which is subsequently input into a machine learning model. An intermediate fusion that integrates feature representations gained from one modality at intermediate layers of a neural network with feature representations learned from other modalities is referred to as joint fusion. Late fusion involves decision-level fusion, in which a distinct model is trained for each modality, and the predictions of all models are subsequently integrated to create a final decision.The final level in the framework is the Knowledge level, where we have the final results of the diagnosis.

Numerous studies have delved into the fusion of diverse modalities for AD diagnosis. Notably, Dwivedi et al. [7], Dong et al. [6], Xu [21], Ning [15], Hao [10], and Zhang [22] have introduced methodologies primarily focused on neuroimaging features, particularly utilizing MRI and PET modalities. Similarly, Khvostikov et al. [12], Kang [11], and Aderghal et al. [1] have directed their attention to the fusion of neuroimaging data, specifically from sMRI and DTI scans. In addition to these, Zuo et al. [24] integrated sMRI, PET, and fMRI data, while Choi and Jin [5] used flurodeoxyglucose and florbetapir PET. Peng et al. [16] combined sMRI, PET, and genetic data, and Lee et al. [14] integrated cognitive performance, demographic information, CSF, and MRI imaging data.

In reviewing these studies, it becomes evident that the most frequently fused modalities are MRI and PET, indicating their prominent role in multimodal investigations within this research domain. Various approaches for fusing MRI and PET volumes have been explored in the literature. For example, Song et al. [17] introduced a framework for AD diagnosis using a feature fusion approach to extract semantic information from 3D MRI and PET volumes. They also proposed

an image fusion method that outperformed their initial approach by reducing the number of model parameters through the use of a single composite image, albeit requiring multistep preprocessing. Kong et al. [13] similarly employed an image fusion technique, while Venugopalan et al. [19] utilized 3D CNNs to extract features from MRI and PET data, demonstrating improved performance over traditional fusion methods despite being limited by dataset sizes.

In our study, we develop an automated multimodal system that integrates MRI and PET images at an intermediate fusion level, facilitating the early diagnosis of AD. This fusion method requires minimal preprocessing compared to traditional image fusion techniques. Our approach surpasses previous studies in distinguishing between individuals with AD and CN individuals.

## 2 Methodology

To preserve the modality-specific information for both modalities, we introduce a heuristic intermediate feature fusion framework that can capture the complementary information from PET and MRI modalities independently. The components of our proposed feature fusion framework are illustrated in figure 1. The first level in our framework is to define the modalities to be integrated. Next, we apply preprocessing steps from the fast-proposed pipeline, to both MRI and PET scans separately to prepare the data for the feature extraction step. In the feature extraction step, a 3D pretrained deep learning model is used as a feature extractor for each modality. Subsequently, we employ an intermediate feature fusion approach by leveraging the feature maps extracted from the previous step and processing them for input to the classification network. Finally, a small and simple 3D CNN network is used as a classification network for the effective classification of AD stages.

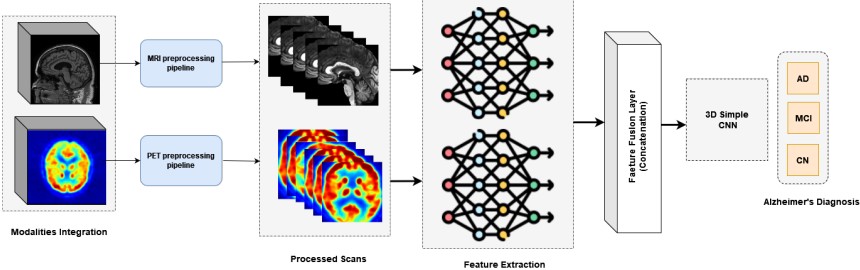

**Fig. 1.** The proposed Intermediate Feature Fusion Framework.

### 2.1 Dataset

Our study concentrates on the ADNI dataset (adni.loni.usc.edu), which is widely used for this problem. We specifically implemented our experiments using the

structural MRI and 18-fluorodeoxyglucose (FDG)-PET modalities, which are commonly employed non-invasive methods for capturing characteristics of brain tissue. We collected 3D data from subjects who underwent scans using both of these modalities.

Firstly, we filtered the subjects to only include those with data available in both PET and MRI during the same visit and scan period. In total, 253 subjects participated in this experiment, contributing to a dataset of 822 scans. We aimed to mitigate the risk of data leakage by considering only the first scans or baseline scans for each subject. This decision ensured an equal number of scans and subjects. However, addressing the challenge of a small dataset size due to the constraint of scans from the same time period, each subject could have 3 to 4 visits in different years or a 6-month gap within the same year. To maintain our principle of avoiding data leakage, we carefully split the data, ensuring that a subject's scans do not appear in different sets but rather all in one place.

The summary of subjects and scans in the dataset is provided in Table 1. All subjects in the dataset belong to either the AD, MCI, or CN groups.

**Table 1.** Summary of participant statistics in the ADNI dataset (MRI and PET).

| Class | Subjects | Scans |
|-------|----------|-------|
| AD | 43 | 117 |
| MCI | 111 | 433 |
| CN | 99 | 272 |
| Total Number of scans = 822 | | |

## 2.2   Data Preprocessing

Both MRI and FDG-PET images in ADNI have undergone various processing stages. Each modality was preprocessed separately. Specifically, the MRI images underwent a series of processing steps, including skull stripping, intensity normalization, uniform resampling to achieve isotropic resolution, 3D cropping to extract only the brain from the black background, resizing, and finally, the application of histogram equalization to enhance the contrast. The preprocessing pipeline that was proposed in [9] was applied here except for the histogram equalization step, which is applied to the scans to enhance the quality and discriminatory power of the images.

Regarding the PET scans preprocessing, the initial FDG-PET scans undergo the following processing steps to ensure consistency in PET data across various systems. Firstly, we converted all the PET files to Neuroimaging Informatics Technology Initiative (NIFTI) format files as all the processed PET images data are in DICOM format. A dicom2nifti python package is used to apply the conversion. Beyond the brain tissue region, PET modality image like MRI exhibits

multiple background areas characterized by pixel values of zeros. We effectively reduce these non-essential background regions to decrease the volume of input data via 3D cropping as in the MRI pipeline. Furthermore, we resized the volumes to be 128×128×128 in size. Finally, histogram equalization is applied to the PET scans. Figure 2 shows the details of the PET preprocessing pipeline and the output of the pipeline.

In this paper, handling multimodal data poses a significant challenge due to the limitation of a small sample size. To address this concern, an essential component of our proposed methodology is the augmentation step. We employ various 3D transformations on both MRI and PET data, including random rotation and flipping.

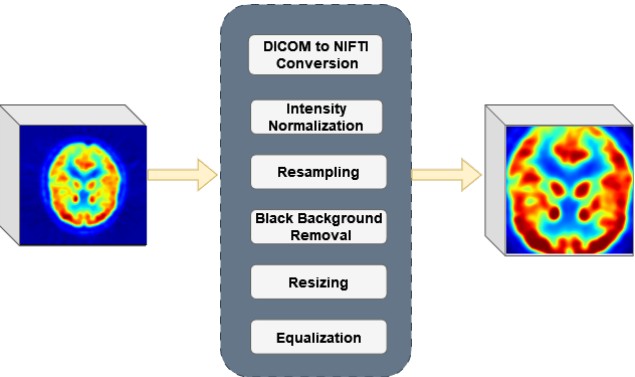

**Fig. 2.** PET Image Processing Method.

### 2.3    Networks Architecture

The effectiveness of the 3D CNN models and transfer learning approach in diagnosing AD, led us to choose them as the optimal starting point for designing our multi-modal framework.

Our proposed multi-modal model architecture is shown in figure 3. The 3D DenseNet201 based transfer learning model is used as a deep feature extractor for the processed images of both modalities separately. After extracting the feature maps from each modality, a concatenation layer is added to the model to fuse those intermediate features and make them ready for the single final network. The last layers in our network form a small and simple 3D CNN. The layers details of the final classification network are also illustrated in figure 3.

## 3    Experiments Setup and Results

In this part of our study, our experiments are organized as follows: initially, two 3D DenseNet201 models are utilized as feature extractors for both MRI and

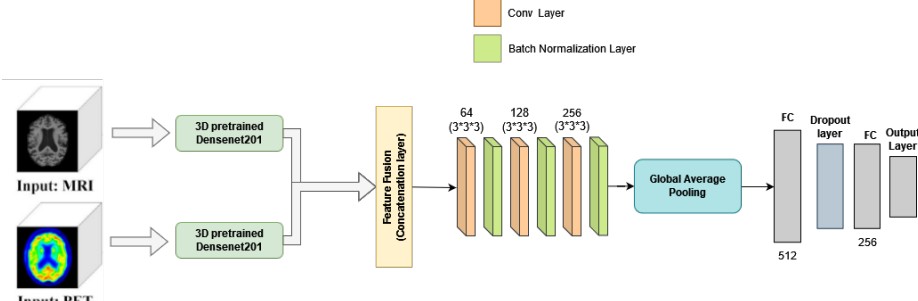

**Fig. 3.** The Proposed Multi-modal Model Architecture.

PET images. Subsequently, we load the weights of each modality independently and incorporate them into the fusion phase. Finally, a straightforward 3D CNN network is applied to the fused features for AD diagnosis. We conduct four classification tasks: AD vs CN, AD vs MCI, and MCI vs CN.

One of the challenges highlighted in the literature is the variability in hyperparameter choices across different studies and experiments. In response to this issue, we employed an open-source hyperparameter optimization framework, Optuna [2]. Optuna is compatible with any machine learning or deep learning framework, offering versatility. Using Optuna's intuitive syntax, we defined the hyperparameter search space and objective function within our existing codebase.

We employed Optuna's automated hyperparameter optimization algorithms, which efficiently explored and evaluated different configurations, enabling us to discover optimal settings for our models. Specifically, we specified the search space for hyperparameters: batch size, learning rate, and input shape by defining their types as categorical, float, and categorical, with possible ranges of [5, 8, 16, 32], [0.000001, 0.0001], and [64, 96, 128] respectively. The framework conducts multiple trials, each representing a unique set of hyperparameter values sampled from the defined search space, and evaluates the objective function for each trial to obtain the corresponding performance metric. Following the optimization process, optuna returns the best set of hyperparameters that led to optimal performance according to the defined objective function. The optimal configuration obtained was [batch size: 16, input size: 128, learning rate: 3.4885205571560794e-05], achieved at trial 9. All experiments were carried out utilizing the TensorFlow deep learning framework in Python. In the training phase of the feature extractors, we employed a total of 200 epochs with a batch size set at 16, aligning with the recommendations derived from the Optuna optimization process. Nevertheless, when training the final CNN network, we encountered hardware constraints, compelling us to decrease the batch size to 5. Adam optimizer is employed with a learning rate that is recommended from optuna algorithm and a ReduceLROnPlateau strategy is utilized here to reduce

the learning rate when the validation loss has stopped improving. According to the final classification network, all the setups have been the same.

In our study, we addressed the challenge of imbalanced classes by implementing oversampling and class weighting strategies during the training of our fusion model. To overcome class imbalance, we applied oversampling to the minority classes using the resample function. This step ensures that each class is represented adequately in the training dataset, preventing the model from being biased towards the majority class. It randomly selects samples with replacements from the provided class indices, effectively duplicating some samples to achieve the desired oversampling. This is done until the size of the minority class matches the size of the majority class, making the class distribution more balanced in the training data. To further mitigate the impact of class imbalance, we computed class weights using the "compute class weight" function from sci-kit-learn. It is utilized to give different weights to different classes during the training of the model. In the experiments, the BinaryFocalCrossentropy loss function is employed, combining the characteristics of both binary cross-entropy (BCE) and focal loss. Binary Cross-Entropy (BCE) serves as the standard loss function for binary classification problems. On the other hand, focal loss is introduced to address class imbalance in binary classification tasks. It achieves this by modulating the cross-entropy loss and down-weighting the contribution of well-classified examples where the predicted probability is high. This adjustment allows the model to prioritize hard-to-classify examples.

We tested the performance of our fusion model through three binary classification tasks to recognize the three AD stages as listed in table 2. The table shows that the best results were obtained for the AD vs CN task, with an AUC score of 97.67%. Table 3 shows the performance of the proposed intermediate feature fusion method compared with the uni-modal methods for the best-performed task (AD vs CN).

**Table 2.** Proposed Feature Fusion Results for 3 classification tasks.

| Task | ACC | BA | AUC | F1-score |
|------|-----|----|----|----------|
| AD vs CN | 95.24 | 95.71 | 97.67 | 93.33 |
| MCI vs CN | 80 | 77.81 | 86.08 | 72.86 |
| AD vs MCI | 75.0 | 74.23 | 80.54 | 73.4 |

**Table 3.** The uni-modal and Proposed Feature Fusion Results for AD vs CN task.

| Metric | MRI | PET | Fusion method |
|--------|-----|-----|---------------|
| ACC | 68.75 | 87.5 | 95.24 |
| AUC | 72.5 | 94.29 | 97.67 |
| BA | 68.67 | 87.71 | 95.71 |
| F1-score | 67.15 | 87.67 | 93.33 |

## 4    Discussion and Conclusion

In this study, our goal was to utilize the power of neuroimaging multimodal data instead of using the unimodal data alone. Table 4 shows the comparison between our proposed algorithm with other studies in the literature through three classification tasks. Our proposed method outperforms the other studies with a superior performance for the AD vs CN task with an ACC= 95.24%.

As introduced in table 4 there are many studies, some of them follow different approaches for fusing between MRI and PET volumes. Song et al. [17] introduced a framework for AD diagnosis with the feature fusion approach to obtain semantic information from the 3D volumes of MRI and PET. The image fusion approach helps in reducing the number of model parameters as a single composite image is used in the network. On the other hand, it requires multistep preprocessing to achieve this fusion. Kong et al. [13] presented also an image fusion method where PET and MRI images are fused and fed into the network. In addition, Venugopalan et al. [19] method suggest that the deep models for integration also show improved performance over traditional feature-level and decision-level integrations. However, their study suffers from having limited dataset sizes.

Regarding data subjects used in this paper, instead of utilizing only the baseline scans, we obtained three to four scans for each subject in different years to overcome the small data sizes as much as possible. In addition, we took into consideration the problem of data leakage that could happen through having multiple scans for each subject, so, we split the data very carefully to ensure that the scans of each subject will not appear in different sets. By utilizing the oversampling and class weighting in our experiments, we got a superior performance of the model and we can see this effect clearly through investigating the metrics especially the f1-scores for each class in different tasks.

Utilizing the 3D augmentation functions was a great addition to our experiments and to the model's performance. Therefore the ordering of those steps in our experiments is as follows, we first applied oversampling to all the training data in each classification task separately by sampling the minor class to have the same number of samples from the major class. Then, the preprocessing is applied to the over-sampled data and finally, the transformation is applied to only the training data.

As a limitation in this study is the challenge of applying the multi-class classification task as dealing with 3D data from MRI and PET scans is very difficult and requires to have high computational resources.

In conclusion, our study presented a comprehensive framework for aiding in the early diagnosis of Alzheimer's disease through a focus on neuroimaging features. We specifically chose to focus on fusing neuroimaging features by combining 3D MRI scans with 18-FDG PET scans through the introduction of an intermediate feature fusion method. Our proposed fusion framework demonstrated superior results compared to related studies in the literature.

**Table 4.** Comparative performance of our classifiers and competitors.

|  | AD vs CN | MCI vs CN | AD vs MCI |
|---|---|---|---|
| Study | ACC (%) | ACC(%) | ACC(%) |
| Kong et al. (2022) [13] | 93.21 | 86.52 | 85.63 |
| song et al. (2021) [17](feature fusion) | 93.22 | 82.37 | 81.00 |
| song et al. (2021) [17](image fusion) | 94.11 | 88.48 | 84.83 |
| Venugopalan et al. (2021)[19] | 86 | - | - |
| Proposed feature fusion method | **95.24** | 80 | 75 |

**Disclosure of Interests.** The authors have no competing interests to declare that are relevant to the content of this article.

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
