# OpenReview forum: "Deep Feature Fusion Framework for Alzheimer’s Disease Staging using Neuroimaging Modalities"
_MICCAI.org/2024/Workshop/MSB — MICCAI Student Board EMERGE Workshop 2024 Oral_

### Official Review · Reviewer_RzFm · 2024-07-05

**Recommendation:** 4
**Confidence:** 5

**Clarity:**

The paper is generally clear but has some clarity issues that could be addressed with moderate revision

**Feedback:**

I provided constructive feedback in the Weaknesses section and directed the authors to consider those. The only two other items for improvement I would elaborate on are:

- Elaborate on the potential clinical applications of your model. For example, classification results for AD vs. CN seem to differ significantly from the other paired cohorts (~20% drop off in ACC). What are some reasons for this? What could be done to improve further distinguishing those other stages?

- On feature interpretability, implement and discuss methods for interpreting which features are most important for the model's decisions (e.g., SHAP values, integrated gradients). If feature interpretability is impossible, explicitly state this as a limitation and discuss its implications.

**Justification:**

The paper provides very strong results and several techniques to address the problem being addressed; however, there is an evident lack of clarity around the methods used and a lack of interpretability/clinical relevance discussed.

**Reproducibility:**

Some amount of details available for reproducing the main results, and open access details are unclear

**Strengths:**

- The authors use an intermediate feature fusion framework to combine MRI and PET imaging data to diagnose AD. This approach preserves modality-specific information while capturing complementary data from both sources. Compared to traditional image fusion techniques, it requires minimal preprocessing, making it potentially more efficient and generalizable.
- The proposed method achieves very high results, especially for distinguishing between AD and CN participants (95.24% ACC and 97.67% AUC). The reported results outperform previous studies cited in the comparative analysis.
- The authors employed strategies to address a common medical data imbalance challenge, such as using multiple scans per subject over time, implementing oversampling and class weighting, and applying 3D data augmentation.
- The authors use Optuna for hyperparameter tuning, which is more systematic and reproducible than manual tuning.
- The study reports multiple binary classification tasks, comparisons against unimodal approaches, various performance metrics, and comparative analysis against other recent studies in the field.

**Summary:**

The paper implements an intermediate fusion approach on multi-modal MRI and FDG-PET images for selecting features for classifying different stages of Alzheimer's Disease.

**Weaknesses:**

Section 2.1
- What other inclusion/exclusion criteria were used to select samples? For ADNI, similar studies conducted on only MRI + FDG-PET scans available returned 1,000+ participants (see “Multimodal and Multiscale Deep Neural Networks for the Early Diagnosis of Alzheimer’s Disease using structural MR and FDG-PET images”). Even studies with more criteria like MMSE scores return ~400 subjects with MRI and FDG-PET data also included (see “Predicting MCI progression with FDG-PET and cognitive scores: a longitudinal study”).
- The data splitting method could be more specific, e.g., subject-level splitting, including longitudinal scans from 3 - 4 visits per subject. What is the resulting distribution of samples in train/val/test? I suggest that the authors consider a k-fold CV in the future to prevent overfitting.

Section 2.2
- What shape were the MRI images resized to? What are all the transformations used, and what’s the resulting distribution of the dataset?

Section 3
- What was the hardware constraint for scaling down the tuned batch size? How did this affect performance?
- Optuna is not capitalized everywhere.
- For random oversampling, did you use a library to do this? If so, mention and cite it.
- For Tables 2 and 3, were these results from one inference done on a hold-out test set or from a best-performing step? Additionally, were the chosen metrics reported over “macro”, “micro”, “weighted” or “samples” settings? Lastly, I would suggest the authors conduct significance testing (e.g., Wilcoxon signed-rank [non-parametric] or a t-test [parametric]) between the ablation studies reported to support the hypothesis of whether these results are significantly different.
- Provide citations for Optuna, Tensorflow, Adam optimizer, sci-kit learn

Section 4
- I am still unsure about the results displayed here and their interpretations. For one, the study seems to be longitudinal by including more than one type of visit but treats all scans as the same visit. This is very concerning for a medical study as, more than likely, disease progression changes per participant over time. Thus, using their baseline labs across multiple visits is a huge misinterpretation.
- Additionally, it seems the authors used random splitting (on whole subject data), which has been known to skew overfitted results.
- Lastly, and most importantly, I fail to understand what clinical relevance the authors can extrapolate from their approach. What features can be interpreted, and what are those interpretations? If they cannot, that should be listed as a limitation of this work.
- Move Table 4 to page 8 or the Supplementary Materials; having it on page 9 violates the 8-page submission restriction.

---

### Official Review · Reviewer_VvvS · 2024-07-10

**Recommendation:** 4
**Confidence:** 4

**Clarity:**

The paper is generally clear but has some clarity issues that could be addressed with moderate revision

**Feedback:**

- In general, the first few sections of the paper are written a lot better than the later parts. I recommend that the authors polish the later sections of the paper.
- What is the "fast-proposed" pipeline? Where is this introduced?
- Figures 1 and 3 could be merged into one.
- A lot of emphasis is placed on the fact that the method does not require extensive preprocessing of the images, yet how this compares to other methods is not clear.
- Too much space is taken up to explain implementation details that are not a key part of the method, such as the use of Optuna for choosing hyperparameters.

**Justification:**

The paper presents a computationally unexpensive method for distinguishing AD patients from healthy controls. It is easy to follow, and most design choices are well-grounded. However, the results do not show an improvement over existing methods for the more challenging tasks where the intermediate mild cognitive decline is a class. Additionally, there are no comparisons to methods that involve a longitudinal component, leveraging multiple scans from the same individuals.

**Reproducibility:**

Some amount of details available for reproducing the main results, and open access details are unclear

**Strengths:**

- The proposed method is computationally inexpensive, as it utilizes a pre-trained feature encoder and low-resolution CNN.
- Leveraging PET and MRI features obtains better results than when only one modality is observed.
- Good performance is achieved for distinguishing individuals with Alzheimer's disease from healthy controls.
- A good summary is provided in the introduction of feature fusion alternatives.
- In general, good design choices are taken such as

**Summary:**

The authors propose an intermediate feature fusion method that combines PET and MRI for Alzheimer's Disease classification on ADNI. After separate preprocessing of each modality, features are extracted from a pre-trained encoder; and then fused to act as input to a 3D CNN. Augmentation is done on both modalities and minority classes are over-sampled during training.

**Weaknesses:**

- It is unusual that the proposed method achieves the best results for the AD vs. CN task, but worse results than all baselines for the more challenging MCI vs. CN and AD vs. MCI tasks. It is also not clear whether the comparisons were carried out with the same splits and using similar preprocessing and augmentation strategies. Additionally, the authors should explain why they look only at binary classification tasks and not at the 3-class problem.
- Too few details are provided on the feature encoder, which is a key part of the method.
- No central tendency and deviation are reported for cross-validation runs or experiments using different seeds. Additionally, no statistical tests validate that the performance gain obtained by the method on AD vs. CN is statistically significant.
- Big sections of text are dedicated to standard processes such as keeping scans from the same patient on the same split and performing hyperparameter selection.

---

### Official Review · Reviewer_DqzB · 2024-07-10

**Recommendation:** 5
**Confidence:** 4

**Clarity:**

The paper is clear and well-written, with minor areas for improvement in clarity

**Feedback:**

1. Consider expanding the dataset or exploring techniques to better handle limited data sizes.

2. Investigate ways to implement multi-class classification, perhaps through more efficient model architectures or distributed computing.

3. Provide a more detailed comparison with other fusion techniques to better highlight the advantages of the proposed intermediate fusion approach.

4. Explore techniques for visualizing or interpreting the features learned from each modality to provide insights into the model's decision-making process.

5. Consider validating the model on an external dataset to further demonstrate its generalizability.

6. Discuss potential clinical implications and how this method could be integrated into real-world diagnostic processes.

**Justification:**

Major factors leading to this score:  Novel and effective approach: The paper presents an innovative intermediate feature fusion framework for multimodal neuroimaging data (MRI and PET). This approach preserves modality-specific information while capturing complementary features, which is a significant advancement in the field of Alzheimer's disease diagnosis. Practical relevance, and addressing data leakage.

**Reproducibility:**

Sufficient amount of details available for reproducing the main results, but open access is not provided to source code and/or data

**Strengths:**

1. Innovative approach using intermediate feature fusion of MRI and PET data, which preserves modality-specific information.

2. Comprehensive experimental setup, including data preprocessing, augmentation, and handling of class imbalance.

3. Strong performance results, outperforming several previous studies on AD vs CN classification.

4. Use of the Optuna framework for hyperparameter optimization, addressing variability issues in prior work.

5. Careful consideration of data leakage issues when using multiple scans per subject.

**Summary:**

The paper presents a novel approach for early diagnosis of Alzheimer's Disease (AD) using deep learning and multimodal neuroimaging data. The key contributions and summary of the paper include:  Development of an automated multimodal system that integrates MRI and PET images using an intermediate feature fusion framework. This approach aims to capture complementary information from both modalities while preserving modality-specific features. Implementation of a 3D DenseNet201-based transfer learning model as a deep feature extractor for both MRI and PET images, followed by a simple 3D CNN for final classification. Addressing challenges in multimodal data analysis, including data preprocessing, augmentation, and handling of class imbalance through oversampling and class weighting strategies. Utilization of the Optuna framework for hyperparameter optimization, addressing the issue of variability in hyperparameter choices across different studies. Achieving state-of-the-art performance in distinguishing between AD and cognitively normal (CN) individuals, with an AUC score of 97.67% and an accuracy of 95.24%. Comprehensive evaluation of the proposed method on multiple classification tasks (AD vs CN, MCI vs CN, AD vs MCI) and comparison with existing studies in the literature.  The paper contributes to the field of AD diagnosis by demonstrating the effectiveness of multimodal neuroimaging analysis and proposing a fusion method that outperforms previous unimodal and multimodal approaches. This work has potential implications for improving early diagnosis and management of Alzheimer's Disease in clinical settings.

**Weaknesses:**

1. Limited dataset size, despite efforts to include multiple scans per subject.

2. Inability to perform multi-class classification due to computational constraints.

3. Lack of detailed comparison of the proposed method against other fusion techniques (e.g., early fusion, late fusion).

4. Limited exploration of the interpretability of the learned features from each modality.

---

### Official Review · Reviewer_fx3h · 2024-07-11

**Recommendation:** 3
**Confidence:** 4

**Clarity:**

The paper is generally clear but has some clarity issues that could be addressed with moderate revision

**Feedback:**

I am recommending rejection for this paper. My primary concerns are the lack of comparison with existing methods and the quality of writing. The paper fails to provide a comprehensive comparison with state-of-the-art approaches, particularly recent transformer-based models for Alzheimer's Disease detection, and does not adequately differentiate its approach from several similar multimodal methods using MRI and PET fusion. This omission makes it difficult to assess the true impact and novelty of the proposed method.

**Justification:**

Lack of comparison with existing methods and paper writing.

**Reproducibility:**

Some amount of details available for reproducing the main results, and open access details are unclear

**Strengths:**

The use of both MRI and PET data addresses the limitations of single-modality approaches, capturing a more comprehensive picture of AD pathology. The authors attempt to address common issues in the field, such as small dataset sizes and class imbalance, and their approach requires minimal preprocessing compared to traditional image fusion techniques, simplifying the workflow and reducing computational overhead. The paper also reports impressive performance metrics, including an AUC of 97.67% and an accuracy of 95.24%, indicating the effectiveness of their method.

**Summary:**

This paper presents a deep feature fusion framework for Alzheimer's Disease (AD) staging using neuroimaging modalities, specifically MRI and PET scans. The authors propose an automated multimodal system that integrates MRI and PET images at an intermediate fusion level for early AD diagnosis. Their method achieves high performance in distinguishing between AD and cognitively normal (CN) individuals, with an AUC score of 97.67% and an accuracy of 95.24%.

**Weaknesses:**

A significant weakness of this paper is the lack of comprehensive comparison with state-of-the-art methods, particularly recent transformer-based approaches for Alzheimer's Disease detection. This omission limits the reader's ability to gauge the true impact and advancement of the proposed method within the current research landscape. Moreover, the authors do not adequately differentiate their work from several existing multimodal methods that also use MRI and PET fusion for AD diagnosis [1,2,3,4]. Additionally the grammar can be significantly improved. The paper uses both "multi-modal" and "multimodal" throughout the text. Consistency should be maintained. Throughout the paper, there are instances of missing articles (a, an, the) and incorrect verb tenses that should be corrected.

[1] - Transformer-Based Multimodal Fusion for Early Diagnosis of Alzheimer's Disease Using Structural MRI And PET
[2] - Automated detection of Alzheimer’s disease: a multi-modal approach with 3D MRI and amyloid PET
[3] - Alzheimer’s Disease Detection from Fused PET and MRI Modalities Using an Ensemble Classifier
[4] - An Effective Multimodal Image Fusion Method Using MRI and PET for Alzheimer's Disease Diagnosis

---

### Meta-Review · Area_Chair_nPi6 · 2024-07-15

**Recommendation:** Accept (Oral)
**Confidence:** 5

**Metareview:**

The paper presents a strong approach with promising results, particularly in the use of intermediate feature fusion for Alzheimer's Disease classification using MRI and PET data. However, acceptance is conditional on ensuring Table 4 is within the 8-page limit for submission version and addressing the minor change for specific details and citations requested by Reviewer RzFm and VvvS in the camera-ready version.

---

### Decision · Program_Chairs · 2024-07-16

**Decision:**

Accept (Oral)

**Comment:**

The work is promising and is accepted with the condition of minor revisions for the camera-ready version.